# The PYY/Y2R-Deficient Mouse Responds Normally to High-Fat Diet and Gastric Bypass Surgery

**DOI:** 10.3390/nu11030585

**Published:** 2019-03-10

**Authors:** Brandon Boland, Michael B. Mumphrey, Zheng Hao, Benji Gill, R. Leigh Townsend, Sangho Yu, Heike Münzberg, Christopher D. Morrison, James L. Trevaskis, Hans-Rudolf Berthoud

**Affiliations:** 1Cardiovascular, Renal & Metabolic Diseases, MedImmune, Gaithersburg, MD 20878, USA; bolandb@MedImmune.com (B.B.); Gillb@MedImmune.com (B.G.); jimmytrevaskis@gmail.com (J.L.T.); 2Neurobiology of Nutrition & Metabolism Department, Pennington Biomedical Research Center, Louisiana State University System, Baton Rouge, LA 70808, USA; mumphrey@umich.edu (M.B.M.); haozhg@outlook.com (Z.H.); townserl@pbrc.edu (R.L.T.); sangho.yu@pbrc.edu (S.Y.); heike.munzberg@pbrc.edu (H.M.); christopher.morrison@pbrc.edu (C.D.M.)

**Keywords:** obesity, diabetes, body weight, body composition, glucose tolerance, insulin tolerance, incretin, energy expenditure

## Abstract

Background/Goals: The gut hormone peptide YY (PYY) secreted from intestinal L-cells has been implicated in the mechanisms of satiation via Y2-receptor (Y2R) signaling in the brain and periphery and is a major candidate for mediating the beneficial effects of bariatric surgery on appetite and body weight. Methods: Here we assessed the role of Y2R signaling in the response to low- and high-fat diets and its role in the effects of Roux-en-Y gastric bypass (RYGB) surgery on body weight, body composition, food intake, energy expenditure and glucose handling, in global Y2R-deficient (Y2RKO) and wildtype (WT) mice made obese on high-fat diet. Results: Both male and female Y2RKO mice responded normally to low- and high-fat diet in terms of body weight, body composition, fasting levels of glucose and insulin, as well as glucose and insulin tolerance for up to 30 weeks of age. Contrary to expectations, obese Y2RKO mice also responded similarly to RYGB compared to WT mice for up to 20 weeks after surgery, with initial hypophagia, sustained body weight loss, and significant improvements in fasting insulin, glucose tolerance, insulin resistance (HOMA-IR), and liver weight compared to sham-operated mice. Furthermore, non-surgical Y2RKO mice weight-matched to RYGB showed the same improvements in glycemic control as Y2RKO mice with RYGB that were similar to WT mice. Conclusions: PYY signaling through Y2R is not required for the normal appetite-suppressing and body weight-lowering effects of RYGB in this global knockout mouse model. Potential compensatory adaptations of PYY signaling through other receptor subtypes or other gut satiety hormones such as glucagon-like peptide-1 (GLP-1) remain to be investigated.

## 1. Introduction

Roux-en-Y gastric bypass surgery and vertical sleeve gastrectomy are the most effective tools to treat severe obesity and its many comorbidities such as diabetes, cardiovascular disease, and cancer [1,2,3,4]. However, there are significant barriers preventing bariatric surgeries from being performed on a much larger scale, including irreversibility of the invasive surgery, complications [5], weight regain [6], loss of bone mineral density [7], increased alcohol and drug abuse [8], high initial cost, and lack of trained surgeons. Therefore, understanding the mechanism(s) responsible for the greatly beneficial effects of bariatric surgeries could accelerate the development of non-surgical, less invasive alternatives or improved surgical techniques.

Altered secretion and circulating levels of key gut hormones have been proposed as one of the major mechanisms driving metabolic disease resolution following RYGB and other bariatric surgeries [9,10]. Specifically, postprandial circulating levels of both GLP-1 and PYY show a rapid and sustained increase after both RYGB and vertical sleeve gastrectomy (VSG) in humans [11,12,13,14,15,16,17], as well as in rodent models [18,19]. A large body of literature has demonstrated the ability of these gut hormones and stable receptor agonists to suppress food intake and lower body weight [20,21,22,23,24,25] and see [26], for recent review.

Furthermore, acute administration of octreotide, a non-specific inhibitor of gut hormone secretion, 3 months after RYGB in humans increased food intake, suggesting that increased levels of satiety hormones are involved in food intake suppression after bariatric surgery [27]. Also, activity in reward-related brain areas in response to viewing palatable food pictures that was reduced after RYGB was restored with GLP-1 receptor blockade [28], suggesting that increased GLP-1 receptor signaling to the brain may be important for the decreased appetite and weight loss after RYGB. However, in three independent animal studies, long-term interference with GLP-1 signaling in whole body GLP-1 receptor-deficient mice was unable to change the ability of RYGB [29,30] or VSG [31] to reduce body weight and improve glycemic control of both surgeries, suggesting that increased circulating PYY may be more important than GLP-1 for the long-term beneficial effects of RYGB and VSG. Indeed, agonism of the PYY Y2R with PYY_3-36_ or more selective Y2R agonism with modified peptide analogs of PYY_3-36_ produced robust weight loss in obese rodents, replicating some of the benefits of RYGB/VSG [32,33].

The longer-term effects of interference with PYY/Y2R-signaling after gastric bypass surgery have been tested in two rodent models. Entero-gastro anastomosis with pyloric ligature produced less body weight loss over 10 days in PYY-deficient mice compared to WT mice [34]. In contrast, RYGB-induced weight loss was not reversed by intracranial PYY/Y2R blockade for 2 weeks in a rat model [30]. The failure to see an effect with central blockade of the Y2R may indicate the importance of peripheral actions of PYY such as promotion of GLP-1 secretion [35].

The aim of this study was to generate and metabolically characterize a whole body Y2R-knockout (Y2RKO) mouse and to test the role of Y2R-deficiency in the long-term beneficial effects of RYGB on body weight and glycemic control.

## 2. Materials and Methods

### 2.1. Animals and Diets

Animal studies were conducted at MedImmune (Gaithersburg, MD, USA), according to protocols reviewed and approved by the Institutional Animal Care and Use Committee (IACUC) at MedImmune in line with the AstraZeneca Animal Welfare and Bioethics policies, or at the Pennington Biomedical Research Center under approved IACUC protocols. For the research conducted at MedImmune: MI-16-0034; For the research conducted at The Pennington Biomedical Research Center: PBRC-P-949.

Constitutive Y2R-knockout mice (C57BL/6NTac-*Npy2r^em3978Tac^*; Y2RKO) were generated at Taconic (San Diego, CA, USA) by embryonic pronuclear microinjection of Cas9 mRNA with a proximal and distal guide RNA (gRNA) targeting regions either side of exon 2 which contains the entire open reading frame of *Npy2r* (Ensemble gene ID: ENSMUST00000029633). Cas9-mediated gene editing resulted in complete excision of the open reading frame, part of the 5′ untranslated region and all of the 3′ untranslated region. The proximal gRNA spacer sequence was ACATAAGTCGA-TTAACAACT and distal gRNA was CACTTTTCCCACGAGTGTTGT. PCR products from 6 putative founders were subcloned and confirmed by Sanger sequencing (not shown). Genotyping was performed using the following primers: oligo 1-F (5′→3′) TTGATCTCACTCATTGTGGAGC and oligo 2-R (5′→3′) CATCAATTGATGAAGATACAGGC to detect the deleted *Npy2r* locus. Oligo 1-F was used with oligo 3-R (5′→3′) TCTACAGTTTGATTCTCATCTGCC to amplify the wildtype (WT) locus. Study animals were obtained from heterozygous breeding pairs housed at Jackson Labs (Bar Harbor, ME, USA) and were housed in standard caging at 22 °C in a 12-h light:12-h dark cycle at standard temperature and humidity conditions with *ad libitum* access to water and food (except where noted).

### 2.2. Experimental Overview

Three cohorts of mice were used in this study. The first cohort consisted of male WT and Y2RKO mice arriving at 10–12 weeks of age. Following one-week of acclimatization, baseline characteristics including %HbA1c were recorded and used to randomize the mice. Mice were then single-housed and transitioned to either a 10% low-fat (LF) diet (Kcal%: Carb, 58; Fat, 13; Prot, 28.5, 5001, Purina LabDiet, Richmond, IN, USA) or a 60% high-fat (HF) diet (Kcal%: Carb, 20; Fat, 60; Prot, 20, Diet D12492, Research Diets, New Brunswick, NJ, USA) to yield four groups: WT LF, WT HF, Y2RKO LF, and Y2RKO HF. Body weight was measured weekly for 6 months. Six-hour fasting insulin and glucose were determined at monthly intervals for 5 months of the study. Body composition was determined at monthly intervals. At month 3, an intraperitoneal glucose tolerance test (1.5 g/kg) was performed following 6 h of fasting. One week later, in one group of WT and Y2RKO mice, a mixed-meal tolerance test (10 uL Ensure^®^/g) following 6 h of fasting; acetaminophen was added to the Ensure^®^ mixture to allow assessment of gastric emptying (10 g/L). One week later, an additional mixed meal tolerance test was performed to assess baseline and meal-stimulated peptide release. Three days later, animals were terminated and body length determined by measuring the length between the nose and anus of the animal. The proximal duodenum was removed and snap frozen for subsequent gene expression analysis.

The second cohort consisted of normal male and female WT mice, mice heterozygous for the Y2R deletion (HET), and mice homozygous for the Y2R deletion (HOM). After one week of acclimatization on chow diet, baseline characteristics (body weight, 6 h fasting insulin and glucose, and body composition) were recorded and used to randomize the mice. Then, mice were single-housed and transitioned to either a LF or HF diet to yield four groups: WT LF, WT HF, HET HF, and HOM HF. Body weight was measured weekly for 6 months. Six-hour fasting insulin and glucose were determined at monthly intervals for the first 3 months of the study period. Body composition was determined at monthly intervals. At month 4, an intraperitoneal glucose tolerance test (1.5 g/kg) was performed following 6 h of fasting. At month 6, an intraperitoneal insulin tolerance test (1.0 U/kg) was performed following 6 h of fasting. 

A third cohort consisting of 30 male WT and 26 male Y2RKO mice arrived at the Pennington Biomedical Research Center at 10–12 weeks of age. Starting at about 6 weeks of age, some mice were exposed to HF diet or a two-choice diet consisting of HF and LF for the duration of the experiment, except for periods in the metabolic chamber when they were exposed to only HF diet. The rationale for the two-choice diet was two-fold: one, it better mimics the human situation; secondly, we found that mice eat relatively more chow immediately after RYGB to maintain motility of the small gastric pouch. Mice were kept individually on corncob bedding except for the periods of food intake measurements, when they were on grid floors. All animals were kept in climate-controlled rooms at a slightly elevated room temperature of 23–24 °C and a 12/12 h light-dark cycle (lights on at 07:00), except for periods near thermo-neutrality (29 °C) in metabolic chambers.

Mice of each genotype were then stratified into 3 groups, Roux-en-Y gastric bypass surgery (RYGB, *n* = 10/11), sham surgery (Sham, *n* = 10/11), weight-matched to RYGB by calorie-restriction (WM, *n* = 6/8). Body weight was measured before and after surgery every 2–3 days for a total of 20 weeks when the animals were euthanized and tissues harvested. Body composition was measured with NMR before and every 2 weeks after surgery. Food intake was measured before surgery, and on days 1–10 and 60–74 after surgery. Energy expenditure was measured at 2–3 weeks and 16–17 weeks after surgery. Glucose tolerance was measured at 4 weeks and insulin tolerance at 6 weeks after surgery. No other invasive tests, such as large volume blood sampling, were performed so as not to perturb body weight. All mice used for the study were genotyped at the end of the experiment to verify their WT or Y2RKO genotype (Appendix A).

### 2.3. Plasma Parameters and Tolerance Tests in Cohorts 1 and 2

#### 2.3.1. Fasting Blood Glucose and Insulin

For the six-hour fasting blood glucose values in all cohorts and the blood glucose values indicated for the intraperitoneal glucose tolerance tests (GTTs) and insulin tolerance tests (ITTs), a glucometer was used (Ascensia Breeze, Bayer, Germany) with tail-vein blood. For all other tests, plasma glucose was determined via colorimetric glucose oxidase kit (Cayman Chemical, Ann Arbor, MI, USA). Six-hour fasting plasma insulin was determined via ELISA (MSD, Rockville, MD, USA). 

#### 2.3.2. Glucose and Insulin Tolerance Tests

Intraperitoneal glucose tolerance tests were performed at either month 3 or 4 in all cohorts, as indicated. Mice were 6 h fasted and injected intraperitoneally with 1.5 g/kg glucose in saline. Blood glucose was determined at 0, 15, 30, 60, 120, and 180 min for cohort 1; 0, 15, 30, 60, and 120 min for cohort 2; and 0, 15, 30, 60, 90, and 120 min for cohort 3. In Appendix A, some values in the HF groups exceeded the glucometer threshold and were removed from the analysis. Intraperitoneal insulin tolerance tests were performed at month 6 in cohort 2. Mice were fasted for 6 h and injected intraperitoneally with 1.0 U/kg insulin. Blood glucose was determined at 0, 15, 30, 60, and 120 min. In Appendix A, 4 animals in the WT LF group required rescue from hypoglycemia at 30 min, and one animal at 60 min.

#### 2.3.3. Mixed Meal Tolerance Test

For determination of circulating metabolic plasma peptide levels shown in Appendix A, the animals were six-hour fasted, retro-orbitally (RO) bled, gavaged with 10 uL/g Ensure^®^, and then RO bled again at minute 30. The Milliplex MAP mouse metabolic hormone magnetic bead panel was used to assess all peptides (Kenilworth, NJ, USA), except for GLP-2, which was determined using the ALPCO mouse GLP-2 ELISA (Salem, NH, USA).

### 2.4. Measurements in Cohort 3 (RYGB Mice)

#### 2.4.1. RYGB, Sham Surgery, and Weight Matching

RYGB was performed according to a protocol described previously [36]. Briefly, in a jejuno-gastric anastomosis, the cut end of the mid jejunum was connected with a very small gastric pouch and the other end of the cut jejunum was anastomosed to the lower jejunum, resulting in a 5–6 cm long Roux limb, a 9–11 cm long biliopancreatic limb, and a 20–25 cm long common limb. Sham surgery consisted of laparotomy only, without transection of jejunum and stomach. Mice weight-matched to the RYGB group were restricted to about 50–70% of the caloric intake of the RYGB group. Pre-weighed amounts of food (Kcal: ~93% high-fat, ~7% chow) were given once per day during the light period.

Two Y2RKO mice subjected to RYGB surgery were euthanized at 6 and 14 days after surgery due to complications with the surgery. One WT mice with RYGB was euthanized 3 weeks after surgery after rapid excessive weight loss and one WT mouse with RYGB died unexpectedly 4 weeks after surgery. These four mice were not included in any analysis.

#### 2.4.2. Measurement of Body Weight, Body Composition, and Food Intake

Body weight was measured every 2–3 days for RYGB and sham mice, and every day for WM mice. Body composition was measured before and every 2 weeks (±4 days) after surgery with a Minispec LF 90 NMR Analyzer (Bruker Corporation, The Woodlands, TX, USA). Adiposity index was defined as fat mass divided by lean mass.

Food intake was measured for 4–7 days before and 4–10 days after surgery, as well as for 10 consecutive days at 9 weeks after surgery. Total food intake in kcal was derived from intake of high-fat diet (5.24 kcal/g) and regular chow diet (3.02 kcal/g) and by taking spillage into account. Chow preference was calculated as the percentage of total food intake in kcals obtained from regular chow diet.

#### 2.4.3. Measurement of Energy Expenditure, Respiratory Exchange Rate (RER), and Locomotor Activity

Energy expenditure, RER, and locomotor activity were measured at the end of the weight loss period (12–25 days after surgery, referred to as 3 weeks) and during the stable weight phase (111–125 days after surgery, referred to as 17 weeks) in metabolic chambers (Phenomaster/Labmaster, TSE Systems, Germany). All mice were first adapted to eating food from hanging baskets in training cages for 4–6 days. Mice that had difficulty eating from hanging baskets were floor-fed. Energy expenditure was measured at two ambient temperatures, normal room temperature at 23 °C for 3 days and near thermoneutrality at 29 °C for 3 days. Mice were adapted for at least one day to each condition before taking measurements. Energy expenditure is reported both unadjusted in kcal/mouse, and adjusted for lean and total body mass. Locomotor activity was measured in numbers of beam breaks in the X and Y planes (7 mm spatial resolution, 10 ms temporal resolution).

#### 2.4.4. Glucose and Insulin Tolerance Tests, Fasting Insulin and Leptin

Glucose tolerance was assessed at 30 ± 4 days (referred to as 4 weeks) after surgery by injecting α-D-glucose (1.5 mg/kg, 30% *w*/*v* in sterile saline, i.p.) and measuring blood glucose from the tail vein before and at 15, 30, 60, and 120 min after injection, with glucose strips and a glucometer (Onetouch Ultra Strips and Onetouch Ultra Glucometer, LifeScan INC, Milpitas, CA, USA). Glucose tolerance tests were conducted between 09:00 and noon, after 3–5 h of food deprivation. Insulin tolerance was assessed at 43 ± 4 days (referred to as 6 weeks) after surgery by injecting insulin (0.6 U/kg in sterile saline, i.p., Novolin R, Novo Nordisk, Bagsvaerd, Denmark) and measuring blood glucose as above. 

#### 2.4.5. Final Plasma and Tissue Harvest

At 143 ± 4 days after surgery mice were food deprived for 3–5 h and euthanized by decapitation. A few drops of trunk blood were collected and blood glucose was immediately tested using glucose strips as above. An additional 500 µL of trunk blood was collected, treated with 83.5 µL EDTA (Sigma, St. Louis, MO, USA) and a protease inhibitor cocktail (1.5 μL of each of the following: Protease inhibitor, Sigma, St. Louis, MO; DPP-IV inhibitor, EMD Millipore, St. Charles, MO; Prefabloc SC, Roche, Indianapolis, IN, USA), and immediately centrifuged at 4 °C and 3000 RPM for 10 min to separate the plasma from the whole blood. Plasma aliquots were frozen in liquid nitrogen and stored at −80 °C prior to processing. Plasma was subjected to ELISA for measurement of insulin concentrations (MADKMAG-71K Milliplex map mouse adipokine magnetic bead panel—endocrine multiplex assay, EMD Millipore, St. Charles, MO, USA).

### 2.5. Statistical Analysis

For high-fat diet growth curves, data normality was determined by the D’Agnostino and Pearson test. Parametric data were analyzed by one-way or two-way ANOVA followed by Tukey’s test. Data are presented as mean ± SEM. All data were analyzed using GraphPad Prism software version 7.02 (San Diego, CA, USA). Statistical significance was set at *p* ≤ 0.05.

Differential changes in weight, fat mass, and adiposity index between RYGB and sham were analyzed with analysis of variance (ANOVA), followed by Students *t*-tests and considered significant at *p* < 0.05. Food intake, energy expenditure, locomotor activity, RER, insulin and glucose tolerance area under the curve (AUC), and fasting plasma assays were analyzed with two-way ANOVA using treatment group and genotype as between subject variables. Benjamini–Hochberg corrected pairwise *t*-tests with false discovery rate set at 0.05 were used for specific group comparisons. All data are reported as mean ± SEM.

## 3. Results

### 3.1. Y2R-Deficient Mice Grow Normally on Regular Chow Diet and Respond Normally to High-Fat Diet

Male WT and homozygous Y2RKO mice at the age of 6 weeks where fed either a low-fat (10% kcal/fat) or a high-fat (60% kcal/fat) diet for 30 weeks. On low-fat diet, Y2RKO mice grew normally but with a tendency to exhibit slightly less weight gain and fat mass gain after about 20 weeks (Figure 1a–c). On high-fat diet, Y2RKO gained body weight, lean mass, and fat mass at similar rates as WT mice, with no significant differences at any time point up to 30 weeks (Figure 1a–d). Although body length at termination was significantly increased in high-fat fed mice, there were no significant differences between genotypes in either low- or high-fat fed mice (Appendix A). Fasting blood glucose was similarly increased by high-fat diet in both WT and Y2RKO mice, with a peak at around 14 weeks and slight moderation thereafter (Figure 1e). Fasting plasma insulin was also similarly increased by high-fat feeding in both genotypes, with a steady increase (Figure 1f). Similarly, intraperitoneal glucose tolerance assessed at 14 weeks was similarly impaired in the high-fat diet groups, with a tendency for Y2RKO mice exhibiting slightly higher impairment compared to WT mice (Figure 1g,h). 

The response to a 60% high-fat diet was also assessed in female and male heterozygous and homozygous Y2RKO mice in a separate cohort. In general, the effects of high-fat diet were very similar, with no major differences between WT and Y2RKO or Y2R heterozygous (HET) mice (Appendix A). Weight gain and body composition changes were similar in Y2R HET and Y2RKO mice compared to WT mice for both males with the typical convex (Appendix A), and females with the typical concave slope of the weight gain curve (Appendix A). Weight gain was mostly due to fat mass gain and this was also similar for the two genotypes (Appendix A).

Fasting blood glucose levels in males were significantly elevated at 60 days on the high-fat diet compared to the low-fat controls in WT, HET and Y2RKO mice (Appendix A), while fasting plasma insulin levels were significantly elevated at 90 days (Appendix A). Fasting blood glucose in females was not significantly elevated except for the Y2RKO mice at 30 days (Appendix A), while fasting plasma insulin was significantly increased to a similar extent in all groups on high-fat diet at 90 days (Appendix A).

In both males and females, glucose tolerance assessed at ~120 days after the start of high-fat diet was significantly impaired to a similar extent in all three groups on high-fat diet compared to the low-fat controls (Appendix A). Finally, insulin’s ability to lower blood glucose, assessed at ~180 days after initiation of high-fat diet, was significantly reduced in all three groups of male mice on high-fat diet, with no difference in genotype (Appendix A). Although there was a similar trend for female mice, only Y2RKO and WT, but not HET mice, on high-fat diet showed a significant deterioration of insulin tolerance (Appendix A).

Fasting and mixed-meal-induced postprandial plasma levels of PYY, GLP-2, resistin, GIP, glucagon, and leptin were not significantly different in male WT and Y2RKO mice on high fat diet (Appendix A).

### 3.2. Similar Effects of RYGB on Body Weight and Body Composition in Y2RKO and WT Mice

Body weight was similarly affected by RYGB and sham surgery in WT and Y2RKO mice (Figure 2a,b). After an initial body weight nadir reached at about 3–4 weeks after RYGB, there was significant weight regain during the remainder of the 19-week observation period for both genotypes. There was a slight tendency for Y2RKO mice with RYGB to regain more body weight during the last 8 weeks, so that the final body weight loss was only 4% below pre-surgical body weight compared to 9% in WT mice with RYGB, but this difference was not statistically significant. In contrast, mice with sham surgery of both genotypes had only a small and transient reduction and then continued to gain body weight. Body weight changes were mostly due to changes in fat mass and there were no significant differences between WT and Y2RKO mice (Figure 2c–e). Interestingly, calorie restriction for weight-matching resulted in significantly greater loss of lean mass and a higher adiposity index compared to RYGB. While RYGB significantly reduced the size of inguinal, mesenteric, perirenal, and retroperitoneal and interscapular brown fat pads compared to sham surgery in both genotypes, it unexpectedly did not decrease gonadal fat (Appendix A). The effects on total fat mass and fad pad weights were generally reflected by leptin levels at termination (Figure 2f). Notably, leptin levels in Y2RKO mice were higher compared to WT mice with RYGB. Although not significant, this difference is consistent with the larger weight regain of Y2RKO mice with RYGB.

### 3.3. Similar Effects of RYGB on Food Intake And Metabolism in Y2RKO and WT Mice

There were no differences in total energy intake between any groups before surgery (Figure 3a,b). After RYGB, food intake was significantly and similarly suppressed in both genotypes for the first 10 days compared to sham surgery and pre-surgical levels. In weight-matched mice, the amount of food necessary to maintain body weight of the respective RYGB groups was significantly less than food intake of RYGB mice for the first 15 days after surgery, and similar for the two genotypes. Preference for chow, which was very low before surgery, significantly increased during the first 10 days after surgery for both genotypes (Figure 3c,d).

Total daily energy expenditure (TEE; ANCOVA-adjusted for body weight) determined 7–8 weeks after surgery at thermoneutrality (29 °C) was significantly higher in mice subjected to RYGB compared with weight-matched mice but was not different compared to sham-operated mice (Figure 4a). Respiratory exchange rate (RER) measured at thermoneutrality, was significantly higher after RYGB compared to weight-matching and sham surgery in Y2RKO mice (Figure 4b). Although the same trend was observed in WT mice, it was not significantly different. There were no significant differences between any groups in locomotor activity assed in the metabolic chambers at thermoneutrality (Figure 4c).

At termination, liver weight was significantly lower after RYGB and weight-matching compared to sham surgery in both genotypes (Appendix A). However, liver weight was significantly lower in Y2RKO mice with sham surgery compared to their WT counterparts. Both heart and kidney weights were significantly lower in weight-matched compared to RYGB and sham mice of both genotypes (Appendix A). Furthermore, wet weight of the small and large intestines was significantly different between sham, RYGB, and weight-matched mice, with RYGB highest and weight-matched lowest, irrespective of genotype (Appendix A).

### 3.4. Similar RYGB-Induced Improvements of Glycemic Control in Y2RKO and WT Mice

Intraperitoneal glucose tolerance assessed 3 weeks after surgery revealed better glucose tolerance in mice with RYGB and weight-matching compared to sham surgery in both genotypes (Figure 5a). However, the difference in AUC between RYGB and sham was only statistically significant in Y2RKO but not in WT mice. Insulin tolerance assessed 6 weeks after surgery showed significantly better ability of insulin to lower blood glucose after RYGB and weight-matching compared to sham surgery in both genotypes (Figure 5b). Furthermore, fasting plasma insulin and HOMA-IR measured at termination was significantly lower after RYGB compared to sham surgery in both genotypes (Figure 6). Although weight-matching resulted in the same significant improvements in WT mice, it did not do so in Y2RKO mice.

## 4. Discussion

Given the strong anorexigenic and body weight lowering effects of PYY and Y2R-agonists [20,32,33,37,38,39], it could be expected that Y2R-deficient mice eat more and gain more weight than WT mice. Previous findings in Y2R-KO mice are inconsistent. On one hand, global Y2R-KO in male mice resulted in slightly higher body weight that normalized after about 20 weeks of age, and decreased locomotor activity [40]. On the other hand, conditional knockout of the Y2R in the hypothalamus resulted in transiently decreased body weight in the face of increase food intake [41]. In our Y2R-KO model, food intake and body weight on regular chow or high-fat diet were similar in both male and female Y2R-KO and WT mice. If anything, body weight gain after 20 weeks on low-fat diet was slightly less in male Y2R-KO mice. Similarly, there were no significant differences between Y2R-KO and WT mice in fasting blood glucose and insulin and in glucose and insulin tolerance. We cannot rule out that compensatory mechanisms during early development may be responsible for the lack of effect in these germline knockout mice, however our data suggest a lack of tonic anorexigenic action mediated via Y2R.

Given the robustly increased secretion and circulating levels of PYY after both RYGB and VSG in humans as well as rodent models [11,13,14,15,16,17], we further hypothesized that at least some of the appetite and body weight-lowering effects of these surgeries are mediated by increased PYY/Y2R signaling and that they would be abolished or attenuated in Y2R-KO mice. However, contrary to our expectations, RYGB was similarly effective in reducing body weight and improving impaired glucose handling in high-fat diet-induced obese Y2R-KO and WT mice, suggesting that neither central nor peripheral PYY/Y2R signaling is required for these beneficial effects of RYGB. Our findings are in contrast to one previous report in mice with duodenal bypass surgery [34], but are consistent with another report in rats with RYGB [30]. The body weight-lowering effect of duodenal bypass surgery over the first 10 days after surgery seen in WT mice was greatly reduced in mice with PYY-deficiency [34]. The different outcome may be due to the different surgery, the short observation period after duodenal bypass, or deficiency in ligand vs. receptor. It is possible that in our Y2R-KO mice, PYY retained its ability to signal through other Y-receptors, such as Y1R, and Y5R, while this was abolished in the PYY-deficient mice. However, Y2R is the preferred receptor for PYY and there is little evidence for anorexigenic and body weight lowering actions of Y1R and Y5R agonism. PYY Y5R may even counteract the anorectic effects mediated by the Y2R [42]. Our findings are, however, consistent with a previous study in rats, in which two weeks of intracranial infusion of the Y2R antagonist BIIE 0246 did not increase food intake and body weight in obese rats that had undergone RYGB four month earlier and had settled at a lower body weight [30]. By infusing the Y2R-antagonist into the brain, that study may have missed important actions of PYY in the periphery. For example, PYY has been shown to stimulate GLP-1 secretion [35], and via Y1R on pancreatic β-cells, insulin secretion [43]. However, the present findings do not suggest that PYY actions through peripheral Y2R is important for the beneficial effects of RYGB.

Y2R is also expressed in a subpopulation of vagal afferents innervating the lower gastrointestinal tract, where it might be involved in vagal afferent satiety signaling [44,45]. Since integrity of vagal afferents innervating the small intestine through the celiac branches are partially required for the beneficial effects of RYGB in rats [46], a case for Y2R on vagal afferents could be made. However, the present results with global Y2R-KO do not support such a role.

Potential compensatory changes in the secretion of gut and other hormones, particularly the L-cell hormones PYY and GLP-1 in our germline Y2RKO mice may be responsible for the negative findings. Although we found similar fasting and postprandial levels of PYY and other hormones in WT and Y2RKO mice on high-fat diet (Appendix A), we did not assess hormone levels after RYGB and this is a limitation of our study. 

## Figures and Tables

**Figure 1 nutrients-11-00585-f001:**
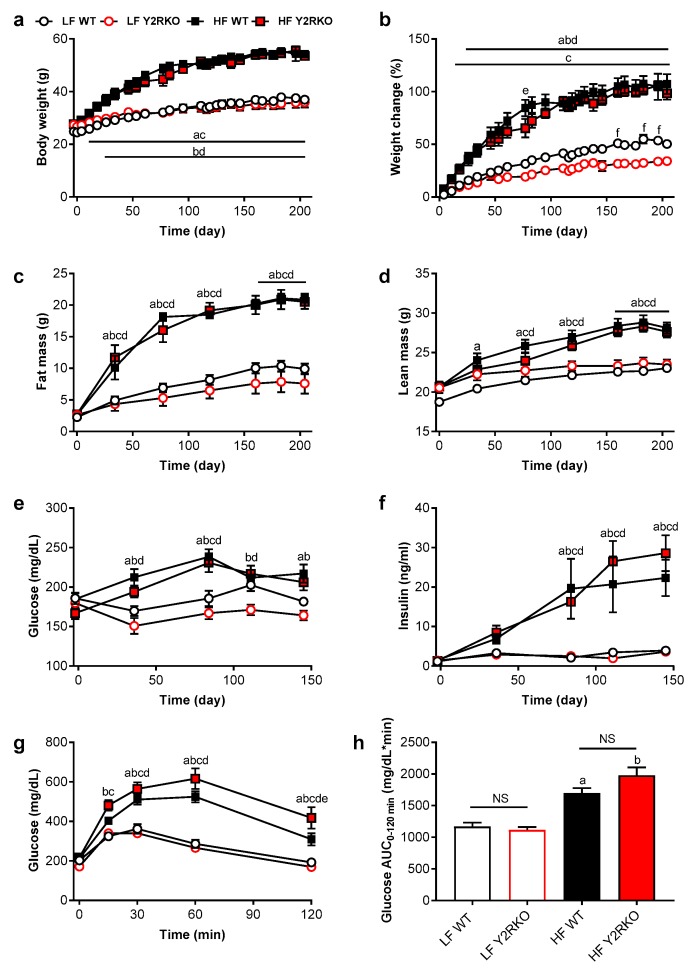
Body weight, body composition, fasting glucose and insulin and glucose tolerance of male wildtype (WT) and Y2R-deficient (Y2RKO) mice on either low-fat (10%; LF) or high-fat (60%; HF) diet. Absolute (**a**) and percent change (**b**) in body weight; total fat mass (**c**) and lean mass (**d**); fasting blood glucose (**e**) and plasma insulin (**f**) and intraperitoneal glucose tolerance test (1.5 g/kg) (**g**) with associated area under the curve analysis (**h**). *n* = 10/group. ^a^
*p* ≤ 0.05 LF WT vs. HF WT, ^b^
*p* ≤ 0.05 LF Y2RKO vs. HF Y2RKO, ^c^
*p* ≤ 0.05 HF WT vs. LF Y2RKO, ^d^
*p* ≤ 0.05 LF WT vs. HF Y2RKO, ^e^
*p* ≤ 0.05 HF WT vs. HF Y2RKO, ^f^
*p* ≤ 0.05 LF WT vs. LF Y2RKO.

**Figure 2 nutrients-11-00585-f002:**
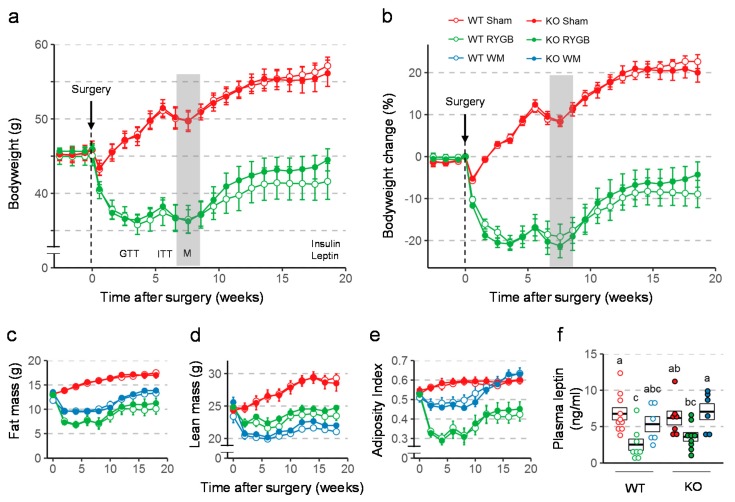
Effect of RYGB or sham surgery, or weight matching to RYGB by caloric restriction (WM) on body weight, body composition, and leptin levels of Y2RKO and WT mice. (**a**,**b**): Effect on absolute body weight and percent body weight change over the 18-week observation period. Body weight of WM mice was, by design, similar to RYGB and is not shown. Timing of glucose tolerance test (GTT), insulin tolerance test (ITT), and metabolic chamber exposure (M, gray bar) are shown in A. (**c**–**e**): Effect on fat mass (**c**), lean mass (**d**), and adiposity index (**e**) over the 18-week observation period. (**f**): Plasma leptin levels at termination of the study. Data in a-e are shown as means ± SEM. Data in f are presented as individual data points over a box showing means ± SEM. Groups that do not share the same letters are significantly different from each other (*p* < 0.05, pairwise *t*-tests with Benjamini–Hochberg correction, false discovery rate (FDR) = 0.05).

**Figure 3 nutrients-11-00585-f003:**
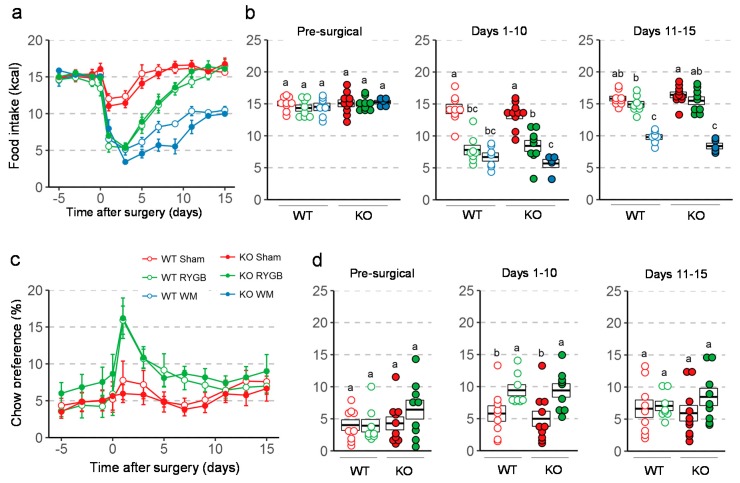
Effect of RYGB, sham surgery, or weight matching to RYGB by caloric restriction (WM) on food intake and diet preference in Y2RKO and WT mice. All mice were on a two-choice diet consisting of high-fat and regular (low-fat) chow. (**a**): Daily food intake before and for 15 days after surgery. (**b**): Average daily food intake over three distinct periods: pre-surgical (5 days before surgery), recovery (days 1–10 after surgery), and stable period (days 11–15 after surgery). (**c**): Daily chow preference before and for 15 days after surgery. Preference is expressed as percent calories obtained from chow diet. WM mice were given a fixed percent of calories from chow so their data is not shown. (**d**): Average daily chow preference over the same periods as in b. Data in a and c are presented as means ± SEM. Data in b and d are presented as individual data points over a box showing means ± SEM. Groups that do not share the same letters are significantly different from each other (*p* < 0.05, pairwise t-tests with Benjamini–Hochberg correction, FDR = 0.05).

**Figure 4 nutrients-11-00585-f004:**
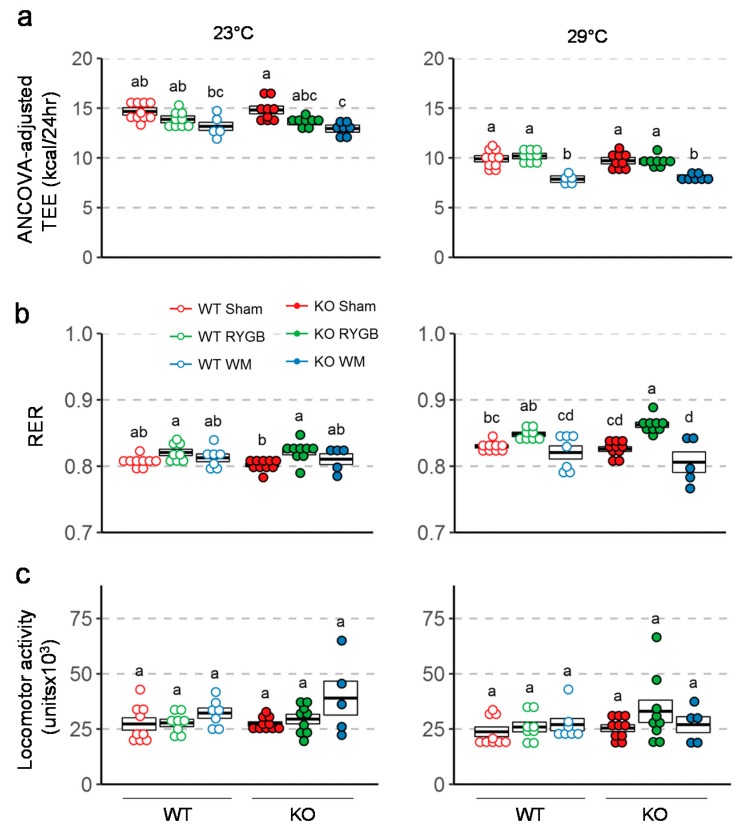
Effect of RYGB, sham surgery, or weight matching to RYGB by caloric restriction (WM) on ANCOVA adjusted energy expenditure using body weight as a covariate (**a**), respiratory exchange ratio (**b**), and locomotor activity (**c**) in Y2RKO and WT mice. Measurements were taken both at room temperature (23 °C) and at thermoneutrality (29 °C) at about 7–8 weeks after surgery. Data are expressed as individual data points over a box showing means ± SEM. Data that do not share the same letters are significantly different from each other (*p* < 0.05, pairwise t-tests with Benjamini–Hochberg correction, FDR = 0.05).

**Figure 5 nutrients-11-00585-f005:**
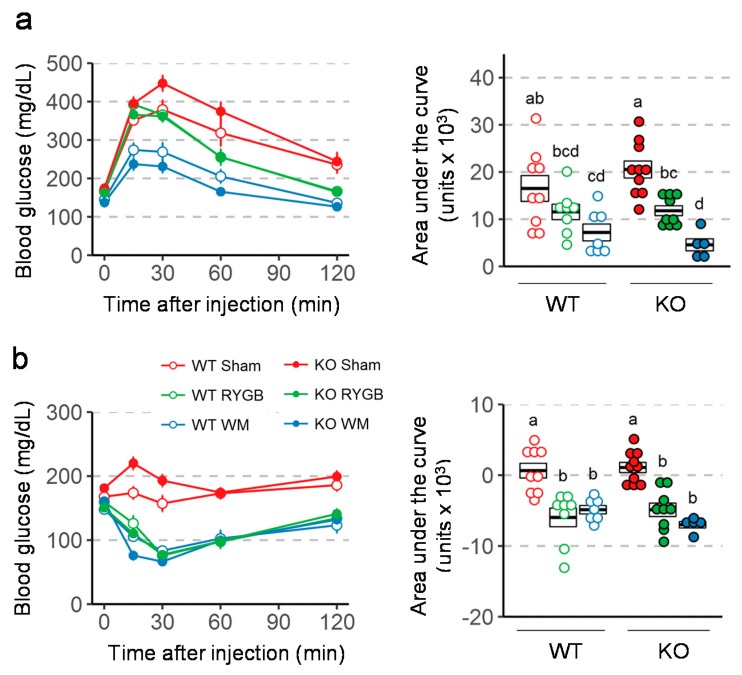
Effect of RYGB, sham surgery, or weight matching to RYGB by caloric restriction (WM) on glucose tolerance (**a**) and Insulin tolerance (**b**) in Y2RKO and WT mice. Data are presented as means ± SEM, or as individual data points over a box showing means ± SEM. Data that do not share the same letters are significantly different from each other (*p* < 0.05, pairwise t-tests with Benjamini–Hochberg correction, FDR = 0.05).

**Figure 6 nutrients-11-00585-f006:**
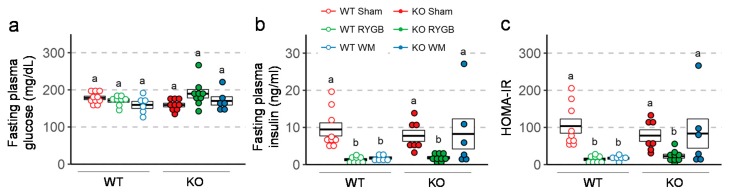
Effect of RYGB, sham surgery, or weight matching to RYGB by caloric restriction (WM) on fasting plasma glucose (**a**), fasting plasma insulin (**b**), and HOMA-IR (**c**), in Y2R-KO and WT mice, as measured at termination of the experiment. Data are presented as individual data points over a box showing means ± SEM. Data that do not share the same letters are significantly different from each other (*p* < 0.05, pairwise *t*-tests with Benjamini–Hochberg correction, FDR = 0.05).

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
