# Peer review of "The PYY/Y2R-Deficient Mouse Responds Normally to High-Fat Diet and Gastric Bypass Surgery"

_nutrients, 2019, doi:10.3390/nu11030585_

Round 1
Reviewer 1 Report
General Comments: In the manuscript entitled "The PYY/Y2R-deficinet mouse responds normally to high-fat diet and gastric bypass surgery" the authors test the hypothesis if PYY through NPY2-R signaling is required for the metabolic effects of high-fat diet and gastric bypass surgery on energy balance. They show that Y2-R deficient mice respond equally to high fat diet in terms of weight gain and glucose regulation. They also show a similar response to gastric bypass in all its effects when compared to wild types.
Comments:
1- As mentioned by the authors in the discussion, there has been previous data in the literature suggesting that PYY deficient mice do not respond to duodenal bypass surgery (a different form of bariatric surgery), while another study in rats showing that central infusion of Y2-recetpro antagonists did not affect the response to RYGB. It may be possible -although less likely- that PYY is acting through a different receptor such as Y1R or Y5R and that's why the Y2R deficient mice still responded to surgery. It might also be, that the Y2R-/- mice lack appropriate endogenous secretion of PYY (for some reason as the receptor was knocked out early and developmental changes and compensations can happen. It would be interesting to examine L-Cell secreted hormones in these Y2-R -/- to examine their response to mixed meal challenge compared to WT AND also show that there is appropriate increase in PYY after RYGB. Otherwise, if these Knock out mice lack adequate PYY secretion-either at baseline or after gastric bypass, this could explain the previous findings in PYY deficient mice .
Author Response
We would like to thank the reviewers and editor for the constructive criticism that allowed us to improve our manuscript.
Reviewer 1:
General Comments: In the manuscript entitled "The PYY/Y2R-deficinet mouse responds normally to high-fat diet and gastric bypass surgery" the authors test the hypothesis if PYY through NPY2-R signaling is required for the metabolic effects of high-fat diet and gastric bypass surgery on energy balance. They show that Y2-R deficient mice respond equally to high fat diet in terms of weight gain and glucose regulation. They also show a similar response to gastric bypass in all its effects when compared to wild types.
Comments: 1- As mentioned by the authors in the discussion, there has been previous data in the literature suggesting that PYY deficient mice do not respond to duodenal bypass surgery (a different form of bariatric surgery), while another study in rats showing that central infusion of Y2-recetpro antagonists did not affect the response to RYGB. It may be possible -although less likely- that PYY is acting through a different receptor such as Y1R or Y5R and that's why the Y2R deficient mice still responded to surgery.
It might also be, that the Y2R-/- mice lack appropriate endogenous secretion of PYY (for some reason as the receptor was knocked out early and developmental changes and compensations can happen. It would be interesting to examine L-Cell secreted hormones in these Y2-R -/- to examine their response to mixed meal challenge compared to WT AND also show that there is appropriate increase in PYY after RYGB. Otherwise, if these Knock out mice lack adequate PYY secretion-either at baseline or after gastric bypass, this could explain the previous findings in PYY deficient mice.
Response: We agree with the reviewer that L-cell hormone secretion could be diminished in the knockout mice. We did measure plasma PYY (but not GLP-1) and a number of other hormones before and after a mixed meal challenge in some wildtype and Y2RKO mice on high fat diet. There were no significant differences in fasting (WT: 107.2 + 8.1; KO: 136.5 + 7.7 ng/ml) and postprandial (WT: 166.5 + 13.8; KO: 145.8 + 28.7 ng/ml) levels of PYY and any other hormones. Unfortunately, we did not make these same measurements after gastric bypass surgery.
We have added this information as a new table to the supplementary information and we have added a couple of sentences to the discussion to make this clear
Reviewer 2:
General Comments: This study investigated the importance of PYY/Y2R receptor signaling in animals exposed to HFD and undergoing gastric bypass surgery. The results provided here indicate that male and female KO animals for the Y2-receptor responded normally to low and high fat diet in terms of body weight, body composition, fasting levels of glucose and insulin, and glucose/insulin tolerance. No difference in response were observed between Y2R-KO animals, wild type animals, and weight-matched animals following RYGB. The conclusion of the authors is that PYY signaling through the Y2receptor does not appear to be required for the normal appetite suppression and body weight-lowering effect of RYGB. Whether compensatory signaling of PYY through other Y receptors occurs in the present Y2R-KO models remains to be addressed.
The study is properly written; the appropriate controls appear to be in place, and the conclusion of the authors is supported by the presented data, although these data are counter-intuitive to our assumption of the role of the gut hormone PYY in the mechanisms of satiation through the Y2Receptor.
Comments: Do the authors have any data/information about the level of adiponectin, leptin, or resistin in the various groups prior to the performance of RYGB procedure? These adipokines could have important contributing regulatory effect on satiation, perhaps enhanced under conditions in which Y2R is removed, thus limiting PYY activity.
Response: As mentioned above, we did measure plasma PYY and a number of other hormones before and after a mixed meal challenge in some wildtype and Y2RKO mice on high fat diet. Unfortunately, adiponectin was not included in our measurements.
Supplementary Table 1
Fasting Postprandial
Peptide | WT | Y2RKO | WT | Y2RKO |
PYY | 107.2 ± 8.1 | 136.5 ± 7.7 | 166.5 + 13.8 | 145.8 + 28.7 |
GLP-2 | 2.42 ± 0.21 | 3.07 ± 0.14 | 2.49 +0.19 | 2.93 + 0.60 |
Resistin | 19549 ± 1597 | 17294 ± 2564 | 23162 + 1861 | 18148 + 2017 |
GIP | 266.4 ± 33.9 | 343.1 ± 101.7 | 365.9 + 82.2 | 233.0 + 60.1 |
Glucagon | 68.3 ± 7.4 | 58.8 ± 8.8 | 90.5 + 8.2 | 71.1 + 7.2 |
Leptin | 15525 ± 2057 | 12567 + 2546 | 18390 + 2346 | 15187 + 3604 |
Reviewer 2 Report
This study investigated the importance of PYY/Y2R receptor signaling in animals exposed to HFD and undergoing gastric bypass surgery. The results provided here indicate that male and female KO animals for the Y2-receptor responded normally to low and high fat diet in terms of body weight, body composition, fasting levels of glucose and insulin, and glucose/insulin tolerance. No difference in response were observed between Y2R-KO animals, wild type animals, and weight-matched animals following RYGB. The conclusion of the authors is that PYY signaling through the Y2receptor does not appear to be required for the normal appetite suppression and body weight-lowering effect of RYGB. Whether compensatory signaling of PYY through other Y receptors occurs in the present Y2R-KO models remains to be addressed.
The study is properly written; the appropriate controls appear to be in place, and the conclusion of the authors is supported by the presented data, although these data are counter-intuitive to our assumption of the role of the gut hormone PYY in the mechanisms of satiation through the Y2Receptor.
Comments:
Do the authors have any data/information about the level of adiponectin, leptin, or resistin in the various groups prior to the performance of RYGB procedure? These adipokines could have important contributing regulatory effect on satiation, perhaps enhanced under conditions in which Y2R is removed, thus limiting PYY activity.
Author Response
We would like to thank the reviewers and editor for the constructive criticism that allowed us to improve our manuscript.
Reviewer 1:
General Comments: In the manuscript entitled "The PYY/Y2R-deficinet mouse responds normally to high-fat diet and gastric bypass surgery" the authors test the hypothesis if PYY through NPY2-R signaling is required for the metabolic effects of high-fat diet and gastric bypass surgery on energy balance. They show that Y2-R deficient mice respond equally to high fat diet in terms of weight gain and glucose regulation. They also show a similar response to gastric bypass in all its effects when compared to wild types.
Comments: 1- As mentioned by the authors in the discussion, there has been previous data in the literature suggesting that PYY deficient mice do not respond to duodenal bypass surgery (a different form of bariatric surgery), while another study in rats showing that central infusion of Y2-recetpro antagonists did not affect the response to RYGB. It may be possible -although less likely- that PYY is acting through a different receptor such as Y1R or Y5R and that's why the Y2R deficient mice still responded to surgery.
It might also be, that the Y2R-/- mice lack appropriate endogenous secretion of PYY (for some reason as the receptor was knocked out early and developmental changes and compensations can happen. It would be interesting to examine L-Cell secreted hormones in these Y2-R -/- to examine their response to mixed meal challenge compared to WT AND also show that there is appropriate increase in PYY after RYGB. Otherwise, if these Knock out mice lack adequate PYY secretion-either at baseline or after gastric bypass, this could explain the previous findings in PYY deficient mice.
Response: We agree with the reviewer that L-cell hormone secretion could be diminished in the knockout mice. We did measure plasma PYY (but not GLP-1) and a number of other hormones before and after a mixed meal challenge in some wildtype and Y2RKO mice on high fat diet. There were no significant differences in fasting (WT: 107.2 + 8.1; KO: 136.5 + 7.7 ng/ml) and postprandial (WT: 166.5 + 13.8; KO: 145.8 + 28.7 ng/ml) levels of PYY and any other hormones. Unfortunately, we did not make these same measurements after gastric bypass surgery.
We have added this information as a new table to the supplementary information and we have added a couple of sentences to the discussion to make this clear.
Reviewer 2:
General Comments: This study investigated the importance of PYY/Y2R receptor signaling in animals exposed to HFD and undergoing gastric bypass surgery. The results provided here indicate that male and female KO animals for the Y2-receptor responded normally to low and high fat diet in terms of body weight, body composition, fasting levels of glucose and insulin, and glucose/insulin tolerance. No difference in response were observed between Y2R-KO animals, wild type animals, and weight-matched animals following RYGB. The conclusion of the authors is that PYY signaling through the Y2receptor does not appear to be required for the normal appetite suppression and body weight-lowering effect of RYGB. Whether compensatory signaling of PYY through other Y receptors occurs in the present Y2R-KO models remains to be addressed.
The study is properly written; the appropriate controls appear to be in place, and the conclusion of the authors is supported by the presented data, although these data are counter-intuitive to our assumption of the role of the gut hormone PYY in the mechanisms of satiation through the Y2Receptor.
Comments: Do the authors have any data/information about the level of adiponectin, leptin, or resistin in the various groups prior to the performance of RYGB procedure? These adipokines could have important contributing regulatory effect on satiation, perhaps enhanced under conditions in which Y2R is removed, thus limiting PYY activity.
Response: As mentioned above, we did measure plasma PYY and a number of other hormones before and after a mixed meal challenge in some wildtype and Y2RKO mice on high fat diet. Unfortunately, adiponectin was not included in our measurements.
Supplementary Table 1
Fasting Postprandial
Peptide | WT | Y2RKO | WT | Y2RKO |
PYY | 107.2 ± 8.1 | 136.5 ± 7.7 | 166.5 + 13.8 | 145.8 + 28.7 |
GLP-2 | 2.42 ± 0.21 | 3.07 ± 0.14 | 2.49 +0.19 | 2.93 + 0.60 |
Resistin | 19549 ± 1597 | 17294 ± 2564 | 23162 + 1861 | 18148 + 2017 |
GIP | 266.4 ± 33.9 | 343.1 ± 101.7 | 365.9 + 82.2 | 233.0 + 60.1 |
Glucagon | 68.3 ± 7.4 | 58.8 ± 8.8 | 90.5 + 8.2 | 71.1 + 7.2 |
Leptin | 15525 ± 2057 | 12567 + 2546 | 18390 + 2346 | 15187 + 3604 |